# Enhanced Cross-Audiovisual Perception in High-Level Martial Arts Routine Athletes Stems from Increased Automatic Processing Capacity

**DOI:** 10.3390/bs15081028

**Published:** 2025-07-29

**Authors:** Xiaohan Wang, Zeshuai Wang, Ya Gao, Wu Jiang, Zikang Meng, Tianxin Gu, Zonghao Zhang, Haoping Yang, Li Luo

**Affiliations:** 1School of Physical Education and Sports Science, Soochow University, Suzhou 215021, China; xiaohanwang0903@126.com (X.W.); zeshuaiwang0616@163.com (Z.W.); jiangjiangibic@163.com (W.J.); mengzikang1017@163.com (Z.M.); gutianxin0811@163.com (T.G.); zhangzonghao@suda.edu.cn (Z.Z.); 2Institute of Psychology, Beijing Sport University, Beijing 100091, China; gaoya520520@bsu.edu.cn

**Keywords:** multisensory channel perception, audiovisual integration, automatic processing, attentional blink, martial arts routine athletes, expert advantage effect

## Abstract

Multisensory integration is crucial for effective cognitive functioning, especially in complex tasks such as those requiring rapid audiovisual information processing. High-level martial arts routine athletes, trained in integrating visual and auditory cues for performance, may exhibit superior abilities in cross-audiovisual integration. This study aimed to explore whether these athletes demonstrate an expert advantage effect in audiovisual integration, particularly focusing on whether this advantage is due to enhanced automatic auditory processing. A total of 165 participants (81 male, 84 female) were included in three experiments. Experiment 1 (n = 63) used a cross-audiovisual Rapid Serial Visual Presentation (RSVP) paradigm to compare the martial arts routine athlete group (n = 31) with a control group (n = 33) in tasks requiring target stimulus identification under audiovisual congruent and incongruent conditions. Experiment 2 (n = 52) manipulated the synchronicity of auditory stimuli to differentiate between audiovisual integration and auditory alerting effects. Experiment 3 (n = 50) combined surprise and post-surprise tests to investigate the role of automatic auditory processing in this expert advantage. Experiment 1 revealed that martial arts routine athletes outperformed the control group, especially in semantically incongruent conditions, with significantly higher accuracy at both lag3 (*p* < 0.001, 95% CI = [0.165, 0.275]) and lag8 (*p* < 0.001, 95% CI = [0.242, 0.435]). Experiment 2 found no significant difference between groups in response to the manipulation of auditory stimulus synchronicity, ruling out an alerting effect. In Experiment 3, martial arts routine athletes demonstrated better performance in reporting unexpected auditory stimuli during the surprise test, indicating enhanced automatic processing capacity. Additionally, a significant improvement in working memory re-selection was observed in the martial arts routine group. The expert advantage effect observed in martial arts routine athletes is attributable to enhanced cross-audiovisual integration, independent of an auditory alerting mechanism. Long-term training improves the efficiency of working memory re-selection and the ability to inhibit conflicting information, suggesting that the expanded capacity for automatic auditory processing underpins their multisensory integration advantage.

## 1. Introduction

The perception of audiovisual information is crucial during high-level martial arts routine movements. Athletes need to collect visual information to maintain the quality of their physical actions, simultaneously gather effective auditory information for movement adjustments, and inhibit irrelevant information such as distractions from outside the arena. This process reflects the attentional allocation resources required by martial arts routine athletes when facing complex motor tasks.

The Rapid Serial Visual Presentation (RSVP) test is commonly used to measure attentional allocation ability. When the time interval between target 1 and target 2 is between 200 and 500 ms, there is an attentional blink (AB) ([20]). AB refers to a brief period after identifying one target during which the detection of a second target (occurring shortly thereafter) is severely impaired. This impairment suggests a temporary depletion or “bottleneck” of attentional resources after processing the first target, leading to a failure in processing subsequent stimuli ([14]). This phenomenon is most common in tasks that require participants to detect specific target items within a rapid sequence of visual stimuli (e.g., letters or shapes).

The attentional bottleneck theory posits that an individual’s attentional allocation capacity within a certain time range is limited, and this resource changes with the “bottleneck” size. This theory explains the fluctuation in attentional limitations within a certain range from a dynamic perspective ([9]; [14]; [19]). When visual stimuli and auditory stimuli appear synchronously, the attentional bottleneck changes, primarily manifested as an enhancement of visual targets ([1]). Multisensory integration theory suggests that cross-audiovisual perceptual enhancement is a result of audiovisual integration, occurring in the early stages of perception ([13]; [30]). Feature-selective attention and audiovisual semantic integration are two brain functions involved in audiovisual object recognition ([13]; [30]). People typically selectively attend to one or more features while ignoring other features of audiovisual objects. Relevant studies indicate that when visual stimuli are congruent with auditory stimuli, it more efficiently promotes visual target recognition ([21]; [25]; [26], [27]); when visual stimuli are incongruent with auditory stimuli, visual stimuli are affected by semantic information ([26], [27]). Other studies suggest that this visual enhancement is an alerting effect induced by auditory stimuli. Related research suggests that other characteristics of auditory information, such as loudness and pitch, additionally activate alertness in attention ([22]; [13]), increasing attentional resources, thereby strengthening the features of visual information. For example, studies have found that even irrelevant or conflicting information features can still improve the recognition of visual object targets ([15]; [20]; [26], [27]).

Attention is generally considered a crucial link for information to enter working memory. Unimportant information is blurred to improve the accuracy and resources for selecting key information ([3]; [13]). The working memory re-selection model suggests that a given stimulus evokes multiple representations in different potential spaces, and these representations focus on different aspects. The memory encoding mechanism can select one or more of these potential spaces for encoding ([7]), which is analogous to reducing the gain associated with other attributes (i.e., lowering the precision of unselected features) ([3]). Working memory then guides attention to re-select already-blurred information. This theory can explain the effect of different sound stimuli on visual information.

Research on martial arts routine athletes has found that their cognitive abilities and audiovisual perception are superior to those of the general population, which is widely attributed to the cognitive gains from exercise experience ([10]; [4]; [17]). Studies have shown that long-term martial arts exercise experience significantly improves cross-audiovisual channel perception and reduces the illusion phenomenon caused by audiovisual incongruence ([10]; [4]; [17]).

The influence of sports experience on the psychological processes of cross-audiovisual perception is not yet clear. How sports experience affects various cognitive processes and improves an individual’s attentional allocation ability in cross-audiovisual channels has not been thoroughly explored. This study, through three series of experiments, investigates whether martial arts routine athletes exhibit an expert advantage effect ([16]) in cross-audiovisual perception tasks, whether this advantage effect is caused by auditory alertness, and whether it is caused by an increase in the automatic processing capacity of auditory information. These experiments are conceptually interconnected: Experiment 1 confirms the presence of an expert advantage (and its robustness to congruency), Experiment 2 isolates the contribution of temporal integration versus simple alerting, and Experiment 3 examines the role of automatic encoding and memory. Together, the three studies build a comprehensive picture of why martial arts athletes may outperform controls in cross-modal tasks, rather than simply demonstrating that they do.

## 2. Experiment 1

### 2.1. Method

Experiment 1 was designed to investigate the presence of an expert advantage effect in cross-audiovisual perception tasks, focusing on whether elite martial arts routine athletes exhibit enhanced perceptual performance relative to non-experts.

**Hypothesis** **1a.**
*High-level martial arts routine athletes perform better than ordinary university students in audiovisual perception tasks, both in congruent and incongruent conditions.*


**Hypothesis** **1b.**
*All groups perform better in audiovisual congruent tasks than in audiovisual incongruent tasks.*


#### 2.1.1. Participants

G*Power 3.1.9.2 was used to estimate the sample size ([5]). With f = 0.25, α = 0.05, and power = 0.8, G*Power suggested a sample size of 34 participants per experiment. In actual recruitment, all research subjects were recruited through online platforms and community posters. All participants provided written informed consent before the experiment, including consent for the publication of anonymized data. The study was approved by the Sports Science Experiment Ethics Committee of Beijing Sport University, 2024349H, and was conducted in accordance with the Declaration of Helsinki. Participants were compensated after completing the experiment.

A total of 173 individuals participated in the experiments. All participants were right-handed, with unimpaired or corrected-to-unimpaired vision and hearing. None had a history of mental illness. Eight participants were excluded due to providing too few correct visual target reports (T1 accuracy below 50%). Ultimately, 165 individuals were included in Experiment 1, Experiment 2, and Experiment 3, as shown in Table 1:

Experiment 1 included 30 high-level martial arts routine athletes (13 males) and 33 university students without any formal athletic training or competitive sports experience (17 males). All participants in the control group were healthy, with unimpaired or corrected-to-unimpaired vision and hearing, and no history of neurological or psychiatric disorders. While they may have engaged in general physical education courses or low-frequency recreational activity (e.g., occasional jogging or non-structured exercise), they did not participate in organized sports clubs, specialized training, or structured athletic programs. Participants with a history of professional sports training, regular participation in competitive sports, or prior martial arts experience were excluded from the control group.

#### 2.1.2. Apparatus, Stimuli, and Experimental Setup

The experiment used a 19-inch monitor (resolution 1280 × 1024, refresh rate 60 Hz) with a gray background (RGB: 196,196,196). Visual stimuli consisted of numbers and letters. Numbers 0–9 were distractors, presented randomly in alternating odd/even order. Capital letters were target stimuli (T1 and T2), with easily confusable letters excluded to ensure experimental accuracy. The final letters E, G, J, U, V, R, and B were used for T1, and K, T, P, A, Z, N, and Y were used for T2.

Auditory stimuli included single-syllable letter sounds that were semantically congruent or incongruent with Target 2 (T2). The volume of auditory stimuli ([25]) was 75 dB, with each single sound lasting 75 ms, followed by a 25 ms interval to ensure clarity and discriminability of the auditory stimuli.

#### 2.1.3. Experimental Procedures and Design

The experimental program was coded using MATLAB 2019a and Psychtoolbox 3.1.2 ([5]) and used for data collection. At the beginning of each trial, a fixation point appeared in the center of the screen for 1000 ms. Subsequently, 17 random visual stimuli were presented sequentially, each consisting of an 83 ms number or letter and a 17 ms blank, totaling 100 ms. Target 1 (T1) was randomly presented at positions 5–9 in the stimulus stream, and Target 2 (T2) was presented at position 1 (lag1 condition), 3 (lag3 condition), or 8 (lag8 condition) after T1. After all visual stimuli were presented, participants were asked to report what T1 and T2 were. As shown in Figure 1. This experiment adopted a 2 (sound condition: semantic congruent, semantic incongruent) × 3 (lag: lag1, lag3, lag8) within-subject factorial design. Each of the six experimental conditions was repeated 40 times, with each participant completing a total of 240 trials.

#### 2.1.4. Statistical Analysis

Correct responses for Target 1 (T1) and correctly reported Target 2 given a correct T1 (T2T1) were recorded accordingly. The T2T1 data were analyzed using a 2 × 2 × 2 repeated-measures ANOVA. When interaction effects reached significance, follow-up simple effect analyses were conducted with Bonferroni correction to control for multiple comparisons. Effect sizes are reported as partial eta squared (η^2^), with thresholds for interpretation set as follows: 0.01 indicates a small effect, 0.06 a medium effect, and 0.14 a large effect. There were no missing data in this experiment. Prior to the ANOVA, assumptions of normality and homogeneity of variance were evaluated using the Shapiro–Wilk test and Levene’s test, respectively, with no violations detected.

### 2.2. Results

In the T1 task, the main effect of auditory congruency was not significant, *F*(1, 64) = 2.65, *p* = 0.109, η^2^ = 0.040, nor was the main effect of martial arts experience, *F*(1, 64) = 1.79, *p* = 0.185, η^2^ = 0.027. In contrast, the T2 task revealed a significant main effect of martial arts experience, *F*(1, 64) = 59.128, *p* < 0.001, η^2^ = 0.480, indicating that martial arts athletes outperformed controls. The main effect of lag was also significant, *F*(2, 128) = 97.103, *p* < 0.001, η^2^ = 0.603, as was the main effect of audiovisual congruency, *F*(1, 64) = 84.215, *p* < 0.001, η^2^ = 0.568. Importantly, there was a significant three-way interaction among lag, audiovisual congruency, and martial arts experience, *F*(2, 128) = 17.356, *p* < 0.001, η^2^ = 0.213. Follow-up comparisons clarified the direction of this expert advantage. Under the lag1 condition, martial arts practitioners performed significantly better than controls in both congruent (*p* < 0.001, 95% CI = [0.118, 0.269]) and incongruent (*p* < 0.001, 95% CI = [0.175, 0.286]) trials. This advantage persisted in lag3 for congruent (*p* < 0.001, 95% CI = [0.089, 0.205]) and incongruent (*p* < 0.001, 95% CI = [0.165, 0.275]) trials. Even at lag8, the martial arts group still outperformed controls in congruent (*p* = 0.019, 95% CI = [0.014, 0.149]) and incongruent (*p* < 0.001, 95% CI = [0.242, 0.435]) conditions (Figure 2).

These findings demonstrate a clear expert advantage effect: high-level martial arts routine athletes consistently outperform ordinary university students in cross-audiovisual perception tasks, regardless of congruency. The lack of a congruency × group interaction within each lag suggests that superior performance is not driven by semantic congruence per se but likely by enhanced audiovisual integration or an alerting effect of auditory stimuli.

## 3. Experiment 2

### 3.1. Method

Experiment 2 aimed to investigate whether the expert advantage effect observed in high-level martial arts routine athletes during cross-audiovisual perception reflects enhanced multisensory integration, thereby elucidating the cognitive mechanisms underlying expert performance.

**Hypothesis** **2a.**
*High-level martial arts routine athletes perform better than ordinary university students in audiovisual perception tasks where audiovisual stimuli appear synchronously and where auditory stimuli appear earlier.*


**Hypothesis** **2b.**
*If an earlier appearance of sound produces a significant advantage, it indicates that this advantage stems from an alerting effect induced by sound.*


#### 3.1.1. Participants

The study ultimately included 25 athletes (13 males) and 27 ordinary university students (13 males). Inclusion criteria were the same as in Experiment 1.

#### 3.1.2. Apparatus, Stimuli, and Experimental Setup

Consistent with Experiment 1.

#### 3.1.3. Experimental Procedures and Design

The procedure was generally consistent with Experiment 1, but sound stimuli were only divided into two temporal conditions: appearing 200 ms earlier or appearing synchronously with T2.

#### 3.1.4. Statistical Analysis

Statistical analysis was generally consistent with Experiment 1. Correctly reported Target 1 was recorded as T1. Correctly reported Target 2, given a correct T1 report, was recorded as T2T1. The T2T1 data were analyzed using a 2 × 2 × 2 repeated-measures ANOVA. If interaction effects were significant, simple effect analysis was performed, with Bonferroni correction.

### 3.2. Results

In the T1 task, the main effect of martial arts experience was significant, *F*(1, 51) = 22.102, *p* < 0.001, η^2^ = 0.302, indicating that the martial arts group outperformed the control group. Follow-up pairwise comparisons revealed that participants in the martial arts group performed significantly better than those in the control group under both auditory conditions: when the auditory stimulus was presented in advance, *F*(1, 51) = 18.668, *p* < 0.001, η^2^ = 0.268, and when it was presented synchronously, *F*(1, 51) = 15.239, *p* < 0.001, η^2^ = 0.230. The main effect of auditory congruency was not significant, *F*(1, 51) = 2.351, *p* = 0.131, η^2^ = 0.044. In the T2T1 task, the main effect of martial arts experience remained significant, *F*(1, 50) = 20.066, *p* < 0.001, η^2^ = 0.286, again demonstrating that martial arts athletes performed better than controls. The main effect of lag was also significant, *F*(2, 49) = 5.429, *p* = 0.007, η^2^ = 0.181. Pairwise comparisons indicated significant decreases in performance as the temporal delay increased, with differences between lag1 and lag8 (*p* = 0.002) and between lag3 and lag8 (*p* = 0.005). The main effect of auditory congruency was not significant, *F*(1, 50) = 1.451, *p* = 0.233, η^2^ = 0.028, and the three-way interaction among martial arts experience, lag, and auditory congruency was also non-significant, *F*(2, 49) = 0.545, *p* = 0.581, η^2^ = 0.022 (Figure 3).

These results suggest no significant difference between audiovisual perception tasks where auditory stimuli precede or coincide with visual stimuli, pointing toward an audiovisual integration effect. Notably, high-level martial arts athletes consistently outperformed ordinary university students in both early-auditory and synchronous conditions. This supports the hypothesis that martial arts experience enhances audiovisual integration ability, as demonstrated in Experiment 1. However, it remains to be clarified whether this advantage stems from more efficient auditory information processing.

## 4. Experiment 3

### 4.1. Method

Experiment 3 examined whether the expert advantage effect in high-level martial arts routine athletes is linked to more efficient automatic processing of auditory information during audiovisual perception, by incorporating surprise and post-surprise trials to probe the temporal dynamics of auditory processing.

**Hypothesis** **3a.**
*High-level martial arts routine athletes perform better than ordinary university students in audiovisual perception tasks involving surprise and post-surprise trials.*


**Hypothesis** **3b.**
*Partial information representation is lost during audiovisual integration, termed “attribute amnesia”.*


#### 4.1.1. Participants

The study ultimately included 23 athletes (15 males) and 26 ordinary university students (14 males). Inclusion criteria were the same as in Experiment 1.

#### 4.1.2. Apparatus, Stimuli, and Experimental Setup

Consistent with Experiments 1 and 2. To avoid ceiling effects, the presentation time for each stimulus was shortened from 83 ms to 67 ms ([20]), while the blank screen duration was increased from 17 ms to 33 ms, maintaining an overall presentation duration of 100 ms.

#### 4.1.3. Experimental Procedures and Design

Consistent with Experiments 1 and 2, but a surprise test was added at the 71st trial. Participants were asked to report the auditory stimulus they heard after reporting T1 and T2, by pressing the corresponding key on the keyboard. All surprise tests were conducted under the lag3 condition and were audiovisual incongruent tasks. After the surprise test, participants completed 10 control trials (post-surprise test), which were identical to the surprise test.

#### 4.1.4. Statistical Analysis

Repeated-Measures Analysis of Variance (ANOVA) was used to analyze T2T1 data in the pre-surprise test, exploring the effects of group, time interval, and auditory stimulus consistency on T2T1. Chi-square test was used to analyze the difference in T3 during the surprise test, comparing the performance of the martial arts routine group and the control group in auditory stimulus recognition. Paired Samples Wilcoxon Signed Rank Test was used to analyze T2T1 and T3 data in the post-surprise test, exploring the impact of the surprise test on subsequent test performance.

### 4.2. Results

T1 results showed no significant difference between the groups. A Welch’s *t*-test comparing the martial arts routine group (M = 0.598, SD = 0.185, n = 26) and the control group (M = 0.545, SD = 0.213, n = 24) yielded *t*(46.5) = 1.33, *p* = 0.189, *d* = 0.37, indicating no significant group difference in T1 performance.

Analysis of T2T1 accuracy revealed a significant main effect of group, *F*(1, 48) = 4.61, *p* = 0.037, η^2^ = 0.09, with the martial arts routine group (M = 0.58, SE = 0.03) performing significantly better than the control group (M = 0.48, SE = 0.03). The main effect of lag was also significant, *F*(1, 48) = 49.69, *p* < 0.001, η^2^ = 0.51, with accuracy at lag3 (M = 0.63) significantly higher than at lag1 (M = 0.43). The group × lag interaction was significant, *F*(1, 48) = 5.21, *p* = 0.027, η^2^ = 0.10. Simple effects analyses showed that although both groups improved from lag1 to lag3 (control: *p* < 0.001; martial arts: *p* = 0.001), the martial arts group’s improvement was greater (Figure 4): at lag3 they outperformed controls (*p* = 0.004), whereas at lag8 no group difference was observed (*p* = 0.621).

In the surprise test, 10 participants (45.5%) in the martial arts routine group correctly reported the auditory stimulus compared to 5 participants (25.0%) in the control group. A chi-square test confirmed this difference, *χ*^2^(1) = 3.84, *p* = 0.05, *φ* = 0.30, indicating a moderate effect size and that the martial arts group outperformed controls in unexpected auditory recognition (Figure 5).

In the post-surprise test, for the martial arts routine group at lag3, the median T3 was 0.5, with an interquartile range of [0.3, 0.8]. For the control group at lag3, the median T3 was 0.4, with an interquartile range of [0.2, 0.7]. Wilcoxon signed-rank test results showed Z = −1.96, *p* = 0.05. This indicates that the difference between the two groups approached statistical significance. The martial arts routine group’s T3 report at lag3 was slightly better than the control group, but the difference did not reach significance. For the martial arts routine group at lag3, the median T3 was 0.6, with an interquartile range of [0.4, 0.9]. For the control group at lag3, the median T3 was 0.5, with an interquartile range of [0.3, 0.8]. Wilcoxon signed-rank test results showed Z = −1.78, *p* = 0.07, indicating that the difference between the two groups did not reach statistical significance. The martial arts routine group’s T3 performance at lag3 was slightly better than the control group, but the difference was not significant (Figure 5).

## 5. General Discussion

Experiment 1 confirmed the existence of an expert group advantage effect in martial arts routine athletes in cross-audiovisual perception tasks, which is consistent with previous research. Experiment 1 also verified that the semantic information of auditory input influences cross-audiovisual integration; when auditory information is congruent with visual information, it further promotes the processing of visual targets ([25], [24]; [26], [27]; [29], [28]). All participants showed a decline in performance under audiovisual incongruent information, especially in the control group, where incongruent semantic information interfered with attentional allocation during the attentional bottleneck period, exacerbating attentional blink. This aligns with the bottleneck theory, which posits that attention is a dynamic process. The decline in visual perception in the control group under audiovisual incongruent information reflects their susceptibility to auditory information interference; even a single instance of semantic incongruence can easily affect participants’ attentional processing. This is consistent with previous research, where semantic conflict with auditory stimuli significantly interferes with target stimuli ([6]; [8]). Martial arts routine athletes can better resist the interference caused by incongruent semantic information. Similar scenarios exist in sports competitions, where martial arts routine athletes need to focus their attention better on their visual targets and movement performance, ignoring irrelevant auditory interference from outside the arena, including semantic information from the audience and irrelevant environmental noise. This aligns with attentional selectivity, as martial arts routine athletes can choose information more efficiently by adjusting their attention ([12]).

Experiment 2 further confirmed that this advantage effect arises from audiovisual integration rather than an alerting effect induced by auditory stimuli. When sounds congruent or incongruent with T2 appeared either earlier or synchronously, neither group showed a significant difference in performance across sound conditions. The loudness and pitch characteristics of the sounds did not elicit additional attentional resources from participants. This is consistent with previous research ([25]; [27]).

Interestingly, at lag8, the control group showed a decline in visual target recognition. This might be due to a resource-saving effect ([11]) exhibited by the control group participants, meaning they relied more on auditory channel perception to some extent, reducing visual perception channel processing resources. This explains the phenomenon in Experiment 1 where the control group was easily affected by single semantic information during attentional processing. Similarly, this phenomenon also indirectly confirms that participants’ cognitive resources are limited ([2]; [13]), and the total allocable cognitive resources of the control group are less than those of martial arts routine athletes, implying that martial arts routine exercise experience impacts an individual’s cognition in terms of overall cognitive capacity.

Furthermore, after ruling out the alerting effect of attention, the decline in visual target recognition at lag8 also reflects that semantic information is more easily processed automatically during audiovisual integration, meaning that the integration of audiovisual semantic information occurs in the early, automatic stage of perception ([11]). In sports, this is exemplified by conflicts between athletes and spectators in arenas often arising because athletes automatically process uncivil verbal information from the audience and store it in working memory, thereby inducing unfavorable emotions.

The T1 and T2T1 results in Experiment 3 were consistent with Experiment 1, showing an expert group advantage effect. After reducing the visual stimulus presentation time, participants’ accuracy appropriately decreased, reducing the ceiling effect. In the surprise test and post-surprise test, martial arts routine athletes showed a higher level of auditory information reporting. In the unexpected surprise trials, Martial arts routine athletes’ efficient reporting reflects their stronger automatic processing ability for semantic components in auditory information and larger resource capacity, requiring no additional attentional resources. Earlier research found that semantic information is easily processed automatically, but due to task effects and attentional selectivity, this irrelevant semantic information was not stored in working memory ([23]; [18]), leading to information loss. In the post-surprise test, generally higher accuracy in reporting T3 validated the working memory re-selection model ([3]). This model suggests that individuals re-adjust attentional resource allocation by familiarizing themselves with task requirements, guiding more perceptual information to be encoded. Similarly, the more efficient performance of martial arts routine athletes also reflects their stronger adaptability and more flexible attentional allocation capabilities.

This study has certain limitations. Firstly, the use of simple stimuli (e.g., digits and letters) may limit ecological validity and contribute to ceiling effects in some participants; future studies should consider sport-specific and more complex stimuli. Secondly, findings based on martial arts routine athletes may not generalize to open-skill sports with higher contextual variability. Thirdly, this study focused on behavioral data; future research could incorporate EEG or fMRI to further explore the neural mechanisms underlying automatic processing. Additionally, increasing sample size and refining experimental design may help optimize the application of findings to athlete training and selection.

## 6. Conclusions

In this study, martial arts routine athletes exhibited an expert advantage effect in cross-audiovisual perception tasks. This expert advantage effect is driven by automated audiovisual integration, meaning martial arts routine athletes possess more efficient audiovisual integration abilities. Specifically, the expert advantage effect in martial arts routine athletes stems from their higher capacity for processing auditory information during the process of integrating audiovisual information.

## Figures and Tables

**Figure 1 behavsci-15-01028-f001:**
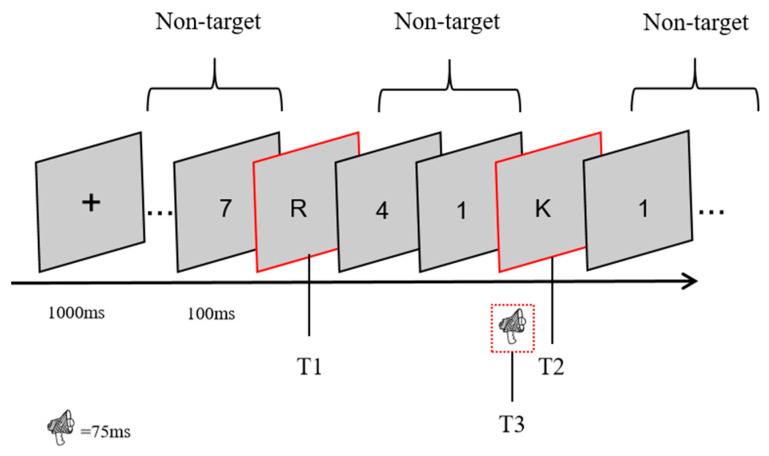
An example trial of Rapid Serial Visual Presentation (RSVP). Participants were asked to report T1 and T2 in all experiments, and T3 in the surprise and post-surprise tests of Experiment 3. T3 stimuli consisted of single-syllable letter sounds that were different from the semantic information of Target 2 (T2), being incongruent. All surprise tests were conducted under the lag3 condition.

**Figure 2 behavsci-15-01028-f002:**
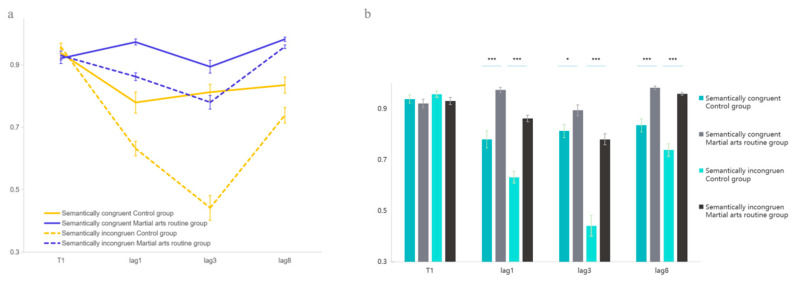
T1 and T2T1 report results in Experiment 1. (**a**) shows the accuracy trends of the two groups in T1 and T2T1 tasks. (**b**) presents the statistical differences between the two groups in T1 and T2T1. Error bars represent ±1 standard error of the mean (SEM). * *p* < 0.05, *** *p* < 0.001.

**Figure 3 behavsci-15-01028-f003:**
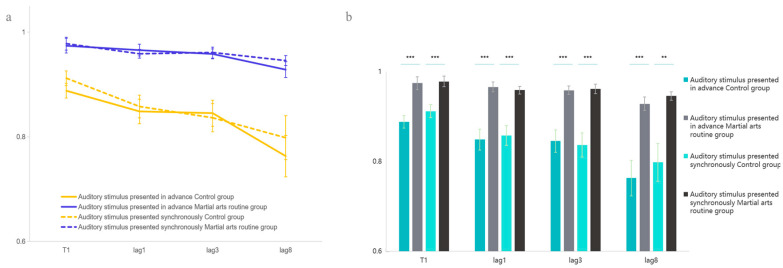
T1 and T2T1 report results in Experiment 2. (**a**) shows the accuracy trends of the two groups in T1 and T2T1 tasks. (**b**) presents the statistical differences between the two groups in T1 and T2T1. Error bars represent ±1 standard error of the mean (SEM). ** *p* < 0.01, *** *p* < 0.001.

**Figure 4 behavsci-15-01028-f004:**
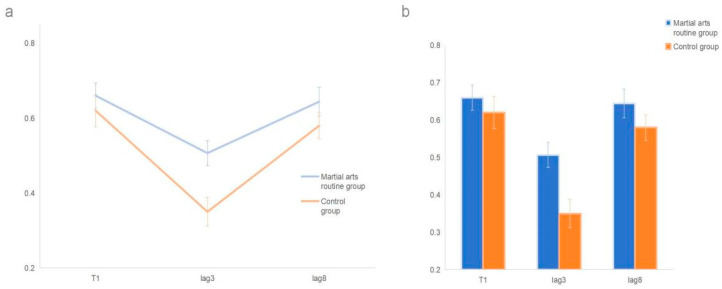
Pre-surprise test T1 and T2T1 report results in Experiment 3. (**a**) shows the accuracy trends of T1 and T2T1 for both groups in the pre-surprise test in Experiment 3. (**b**) shows the statistical differences in T1 and T2T1 for both groups before the surprise test in Experiment 3. Error bars represent ±1 standard error of the mean (SEM).

**Figure 5 behavsci-15-01028-f005:**
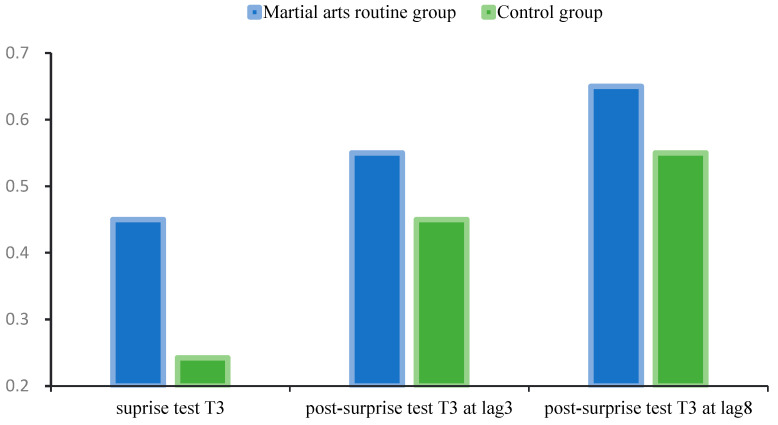
T3 reporting rate in surprise and post-surprise tests. Error bars represent ±1 standard error of the mean (SEM).

**Table 1 behavsci-15-01028-t001:** Basic demographic information of participants.

		Experiment 1		Experiment 2		Experiment 3	
		Martial Arts Routine Group	Control Group	Martial Arts Routine Group	Control Group	Martial Arts Routine Group	Control Group
Age (year)		22.03 ± 1.14	20 ± 1.81	19.95 ± 1.9	20.95 ± 2.1	21.33 ± 2.2	20.31 ± 1.2
Gender	Male	13	17	13	13	15	14
	Female	17	16	12	14	11	10
None			33		27		24
Sport Level	Level 2	23		21		19	
	Level 1	3		3		6	
	Elite	4		1		1	
Total		30	33	25	27	26	24
Avg. Training Duration (year)		8.73 ± 3.2 years		9.72 ± 4.4 years		8.64 ± 3.14 years	

## Data Availability

The original contributions presented in this study are included in the article. Further inquiries can be directed to the corresponding authors.

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
