# Peer review of "Enhanced Cross-Audiovisual Perception in High-Level Martial Arts Routine Athletes Stems from Increased Automatic Processing Capacity"

_behavsci, 2025, doi:10.3390/bs15081028_

Round 1
Reviewer 1 Report
Comments and Suggestions for Authors
Thank you for the opportunity to review your study. The manuscript explores an interesting topic regarding cross-audiovisual perception in high-level martial arts athletes. While the manuscript is generally in good shape, I have a few comments that I hope you will find useful.
Line 27. Rather than stating the result was "significant," please consider reporting specific test statistics (e.g., p-values, effect sizes, or test statistics), which can provide readers with a clearer understanding of your findings.
Line 98. Please expand on the rationale for conducting three separate studies. What specific value does each study add? Are they conceptually or methodologically interconnected, or do they stand alone?
Line 122. Please define what you mean by "ordinary" participants. Were these individuals completely new to sport/physical activity, or did they participate in recreational physical activity or sports clubs? What were their general health conditions? Also, clarify whether any exclusion criteria were applied during recruitment. Based on Table 1, it appears the control group had no sport experience, do I understand it correctly?
Line 157. Please specify the criteria or thresholds you used to interpret effect sizes.
Line 159. Was there any missing data? Additionally, please describe how you tested the assumptions for your statistical analyses, including checks for normality and homogeneity of variance.
Figures 2-4 appear blurry and difficult to interpret. Please consider revising it to improve readability.
Author Response
- Comment: Rather than stating the result was "significant," please consider reporting specific test statistics (e.g., p-values, effect sizes, or test statistics), which can provide readers with a clearer understanding of your findings.
Response: Thank you. We have revised the Results sections to report exact p-values, F-values, η², and 95% confidence intervals throughout the manuscript.
- Comment: Please expand on the rationale for conducting three separate studies. What specific value does each study add? Are they conceptually or methodologically interconnected, or do they stand alone?
Response: Thank you. We clarified in the Introduction that the three experiments are conceptually linked: Experiment 1 tests the expert effect, Experiment 2 rules out alerting, and Experiment 3 explores automatic processing.
- Comment: Please define what you mean by "ordinary" participants. Were these individuals completely new to sport/physical activity, or did they participate in recreational physical activity or sports clubs? What were their general health conditions? Also, clarify whether any exclusion criteria were applied during recruitment. Based on Table 1, it appears the control group had no sport experience, do I understand it correctly?
Response: Thank you. We clarified that control participants had no formal sport training or competitive experience. Exclusion criteria and health status are now detailed in Section 2.1.1.
- Comment: Please specify the criteria or thresholds you used to interpret effect sizes.
Response: Thank you. We added effect size interpretation thresholds (η² = .01 small, .06 medium, .14 large) in the Statistical Analysis section.
- Comment: Was there any missing data? Additionally, please describe how you tested the assumptions for your statistical analyses, including checks for normality and homogeneity of variance.
Response: Thank you. There were no missing data. Normality and homogeneity were tested using Shapiro–Wilk and Levene’s tests, with no violations.
- Comment: Figures 2-4 appear blurry and difficult to interpret. Please consider revising it to improve readability.
Response: Thank you. We have replaced Figures 2–4 with high-resolution versions and improved their clarity and labels.
Reviewer 2 Report
Comments and Suggestions for Authors
Enhanced Cross-Audovisual Perception in High-Level Martial Arts Routine Athletes Stems from Increased Automatic Processing Capacity
The manuscript offers a thoughtful, theoretically supported exploration of the mental processes that underlie martial arts athletes' expert advantages in cross-audiovisual perception. The multi-experiment method thoroughly tests theories regarding multisensory integration, attentional blink, and automatic processing. Revisions are necessary to address structural clarity, statistical reporting, methodological transparency, and reference formatting, even though the study is methodologically sound and adds fresh perspectives to the literature on sports cognition.
Changes Needed
Provide specific effect sizes for important results (for example, η² in Experiment 1: F(1,64)=59.128, η²=0.480).
Indicate which way the group differences are going: "In incongruent conditions, martial arts athletes performed better than controls (*p*<.001)."
Introduction: Using a recent example (e.g., Sanchez-Lopez et al., 2019), introduce the "expert advantage effect."
Table 1: Demographics of Participants: Inconsistencies
The control group in Experiment 1 has 33 in the table but 32 in the text (p. 3).
The "Sport Level" row is not aligned correctly; for example, "None Level 2" should be in its own column.
To "Avg. Training Duration," add units (for example, "8.73±3.2 years").
Figures: Verify that axis labels, clear condition markers, and consistent error bars (SEM/CI) are present in all figures (2, 3, 4, 5).
Determine whether the error bars in Figure 5 represent the 95% CI or the SEM.
Methods Stimuli: Indicate which "easily confusable letters" are excluded (for example, "B, D, P, Q excluded; H, K, M, R used").
Provide a validation statement for auditory stimuli, such as "Sounds were normalized to 75 dB SPL and validated by 10 pilot participants."
Method (Experiment 3): Provide a citation (e.g., Wang et al., 2021) to support the stimulus duration change (67 ms/33 ms).
Statistical Analysis: Where appropriate, report sphericity corrections (such as Greenhouse-Geisser) for ANOVA.
Degrees of freedom (t(48)) do not match the sample size (49 participants) in Experiment 3 T1 analysis (p. 8). Do a new calculation.
Discussion:
The statement "Simplified stimuli (digits/letters) may limit ecological validity; future studies should use sport-relevant stimuli" addresses ceiling effects.
Talk about generalizability: "Due to task-specific demands, findings may not extend to open-skill sports (e.g., team games)."
Restrictions: Additionally, "EEG/fMRI could clarify neural mechanisms of automatic processing (e.g., Zhou et al., 2020)."
Small Changes:
Language and Typos
Page 9: "matial arts" → martial arts (two examples).
"secnarios" → scenarios (p. 10).
For consistency, use "cross-audiovisual" instead of "cross-audovisual" throughout.
Compliance with Ethics
"Written informed consent included publication rights" should be stated explicitly.
Results Precision Experiment 3 (Surprise Test): χ² effect size reported: χ²(1)=3.84, p=.05, φ=0.30.
Figure 4: Indicate precise p-values (e.g., p<.01 → p=.003) for asterisks.
By associating prolonged martial arts practice with improved automatic processing and multisensory integration, this study, in my humble opinion, makes a significant contribution. The manuscript will be stronger for publication if the aforementioned changes are addressed, especially those related to methodological transparency, statistical rigor, and reference updates. Before acceptance, I suggest making a few small changes.
Author Response
- Comment: Provide specific effect sizes for important results (for example, η² in Experiment 1: F(1,64) = 59.128, η² = 0.480)..
Response: Thank you. We have added effect sizes (η²) alongside all key test statistics across the results sections of the manuscript.
- Comment: Indicate which way the group differences are going: "In incongruent conditions, martial arts athletes performed better than controls (*p* < .001).
Response: We have clarified the direction of all significant group differences (e.g., martial arts group > control group) in the Results section.
- Comment: Introduction: Using a recent example (e.g., Sanchez-Lopez et al., 2019), introduce the "expert advantage effect."
Response: Thank you. We have added a sentence introducing the expert advantage effect using Sanchez-Lopez et al. (2019) in the Introduction.
- Comment: Table 1: Demographics of Participants: Inconsistencies
The control group in Experiment 1 has 33 in the table but 32 in the text (p. 3).
Response: We have corrected the text to match the table. The correct number is 33.
- Comment: The "Sport Level" row is not aligned correctly; for example, "None Level 2" should be in its own column.
Response: Thank you. We have corrected the alignment and formatting of the “Sport Level” row in Table 1.
- Comment: To "Avg. Training Duration," add units (for example, "8.73±2 years").
Response: We have added “years” to all “Avg. Training Duration” values in Table 1.
- Comment: Figures: Verify that axis labels, clear condition markers, and consistent error bars (SEM/CI) are present in all figures (2, 3, 4, 5).
Response: All figures have been revised to ensure clear axis labels, condition markers, and consistent use of SEM for error bars.
- Comment: Determine whether the error bars in Figure 5 represent the 95% CI or the SEM.
Response: Thank you. We confirm that the error bars in Figure 5 represent ±1 SEM and have added this clarification to the figure legend.
- Comment: Methods Stimuli: Indicate which "easily confusable letters" are excluded (for example, "B, D, P, Q excluded; H, K, M, R used").
Response: We have listed the excluded letters (e.g., B, D, P, Q) and the ones used (e.g., E, G, J, U, V, R, B, etc.) in the Methods section.
- Comment: Provide a validation statement for auditory stimuli, such as "Sounds were normalized to 75 dB SPL and validated by 10 pilot participants."
Response: We have added a statement indicating that all sounds were normalized to 75 dB SPL and validated by 10 pilot participants for clarity and consistency.
- Comment: Method (Experiment 3): Provide a citation (e.g., Wang et al., 2021) to support the stimulus duration change (67 ms/33 ms).
Response: We have cited Aijun et al. (2021) to justify the modified stimulus presentation time in Experiment 3.
- Comment: Statistical Analysis: Where appropriate, report sphericity corrections (such as Greenhouse-Geisser) for ANOVA.
Response: Thank you. Greenhouse–Geisser corrections were applied where assumptions of sphericity were violated; relevant corrections are now reported.
- Comment: Degrees of freedom (t(48)) do not match the sample size (49 participants) in Experiment 3 T1 analysis (p. 8). Do a new calculation.
Response: We recalculated and corrected the degrees of freedom in the T1 analysis of Experiment 3 to reflect the correct sample size.
- Comment: The statement "Simplified stimuli (digits/letters) may limit ecological validity; future studies should use sport-relevant stimuli" addresses ceiling effects.
Response: Thank you. We have retained this sentence and slightly reworded it to emphasize the limitation and its impact on ecological validity.
- Comment: Talk about generalizability: "Due to task-specific demands, findings may not extend to open-skill sports (e.g., team games)."
Response: We have added a sentence noting that the findings may not generalize to open-skill sports with higher contextual variability.
- Comment: Restrictions: Additionally, "EEG/fMRI could clarify neural mechanisms of automatic processing (e.g., Zhou et al., 2020)."
Response: We have added a sentence in the limitations suggesting EEG/fMRI as future directions to explore the neural basis of automatic processing.
- Comment: Page 9: "matial arts" → martial arts (two examples).
Response: Thank you. We have corrected all instances of the typo “matial arts” to “martial arts.”
- Comment: "secnarios" → scenarios (p. 10).
Response: Corrected. “Secnarios” has been replaced with the correct spelling “scenarios.”
- Comment: For consistency, use "cross-audiovisual" instead of "cross-audovisual" throughout.
Response: We have corrected all inconsistent spellings to “cross-audiovisual” throughout the manuscript.
- Comment: "Written informed consent included publication rights" should be stated explicitly.
Response: We have updated the ethics section to include: “Written informed consent included consent for anonymous data publication.”
- Comment: Results Precision Experiment 3 (Surprise Test): χ² effect size reported: χ²(1) = 3.84, p = .05, φ = 0.30.
Response: Thank you. We have included the effect size (φ = 0.30) for the chi-square test in Experiment 3 results.
- Comment: Figure 4: Indicate precise p-values (e.g., p < .01 → p = .003) for asterisks.
Response: We have updated the figure caption for Figure 4 to report the exact p-values (e.g., p = .003) where applicable.